# Diagnostic Accuracy of Fine-Needle Aspiration Cytology and Core-Needle Biopsy in the Assessment of the Axillary Lymph Nodes in Breast Cancer—A Meta-Analysis

**DOI:** 10.3390/diagnostics10090717

**Published:** 2020-09-18

**Authors:** Jung-Soo Pyo, Jaehag Jung, Seul Gi Lee, Nae-Yu Kim, Dong-Wook Kang

**Affiliations:** 1Department of Pathology, Daejeon Eulji University Hospital, Eulji University School of Medicine, Daejeon 35233, Korea; jspyo@eulji.ac.kr; 2Department of Surgery, Daejeon Eulji University Hospital, Eulji University School of Medicine, Daejeon 35233, Korea; gsjung@eulji.ac.kr (J.J.); seulgi@eulji.ac.kr (S.G.L.); 3Department of Internal Medicine, Daejeon Eulji University Hospital, Eulji University School of Medicine, Daejeon 35233, Korea; naeyu46@eulji.ac.kr; 4Department of Pathology, Chungnam National University Sejong Hospital, 20 Bodeum 7-ro, Sejong 30099, Korea; 5Department of Pathology, Chungnam National University School of Medicine, 266 Munhwa Street, Daejeon 35015, Korea

**Keywords:** fine-needle aspiration cytology, core needle biopsy, axillary lymph node, meta-analysis, diagnostic test accuracy review

## Abstract

Background: The present study aims to evaluate the diagnostic accuracy between ultrasonography-guided fine-needle aspiration cytology (US-FNAC) and core needle biopsy (CNB) of axillary lymph nodes (ALNs) in patients with breast cancer through a meta-analysis and a diagnostic test accuracy (DTA) review. Methods: The present meta-analysis and DTA review included 67 eligible studies. The diagnostic accuracy of various preoperative assessments, including US-FNAC and CNB, was evaluated for ALNs assessments in patients with breast cancer. In addition, a subgroup analysis based on methods of cytologic preparation was performed. In the DTA review, the sensitivity, specificity, diagnostic odds ratio (OR) and area under the curve (AUC) on the summary receiver operating characteristic (SROC) curve were calculated. Results: The diagnostic accuracy of the preoperative assessments of ALNs was 0.850 (95% confidence interval (CI) 0.833–0.866) for patients with breast cancer. The diagnostic accuracy of CNB was significantly higher than that of US-FNAC (0.896, 95% CI 0.844–0.932 vs. 0.844, 95% CI 0.825–0.862; *p* = 0.044 in a meta-regression test). In the subgroup analysis based on cytologic preparation, the diagnosis accuracies were 0.860, 0.861 and 0.859 for the methods of conventional smear, liquid-based preparation and cell block, respectively. In the DTA review, CNB showed higher sensitivity than US-FNAC (0.849 vs. 0.760). However, there was no difference in specificity between US-FNAC and CNB (0.997 vs. 1.000). US-FNAC with liquid-based preparation and CNB showed the highest diagnostic OR and AUC on the SROC, respectively. Conclusion: Both US-FNAC and CNB are useful in preoperative assessments of ALNs in patients with breast cancer. Although the most sensitive test was found to be CNB in this study, there was no difference in specificity between various preoperative evaluations and the application of US-FNAC or CNB may be impacted by various factors.

## 1. Introduction

In breast cancers, the assessment for axillary lymph node (ALN) is important in predicting the patient’s stage and prognosis and in determining treatment guidelines. According to the results of the American College of Surgeons Oncology Group (ACOSOG) Z0011 trial, in the case of clinical node-negative patients, ALN dissection is not performed according to the result of sentinel lymph node biopsy (SLNB) [1]. Preoperative assessments of ALNs in patients with breast cancer mainly include axillary ultrasound sonography (AUS) and/or ultrasonography-guided fine-needle aspiration cytology (US-FNAC) [2]. US-FNAC confirms whether metastatic ALN as suspicious ALNs during AUS. ALN dissection without SLNB is performed in patients with metastatic ALNs detected by US-FNAC. On the other hand, patients found negative using US-FNAC are subjected to the intraoperative SLNB. After AUS and US-FNAC, core needle biopsy (CNB) is recommended as the preoperative assessment. In daily practice, various methods are introduced for cytological preparation, such as conventional smear (CS), liquid-based preparation (LBP) and cell block [3,4,5,6,7,8,9,10,11,12,13,14,15,16,17,18,19,20,21,22,23,24,25,26,27,28,29,30,31,32,33,34,35,36,37,38,39,40,41,42,43,44,45,46,47,48,49,50,51,52,53,54,55,56,57,58,59,60,61,62,63,64,65,66,67,68,69]. In pathological examinations, LBP, which has been widely applied to the screening of uterine cervical lesions, has gradually replaced CS. The diagnostic accuracy can differ between the methods of cytological preparation. However, confirmative information for a comparison of diagnostic accuracy between US-FNAC and CNB is lacking in terms of assessments of ALNs in patients with breast cancer.

In the present study, we investigated and elucidated the diagnostic accuracy of US-FNAC and CNB for ALN assessment in patients with breast cancer. The diagnostic accuracy between US-FNAC and CNB was compared through a meta-regression test. In addition, a diagnostic test accuracy (DTA) review was performed to obtain the pooled sensitivity and specificity, diagnostic odds ratio (OR) and area under the curve (AUC) on the summary receiver operating characteristic (SROC) curve.

## 2. Materials and Methods

### 2.1. Published Study Search and Selection Criteria

Relevant articles were obtained by searching the PubMed database through 31 July 2020. We searched using the following keywords: “((Ultrasound OR US) OR (Ultrasound guided OR US guided) OR (sonography OR sonography guided)) AND ((FNA OR Fine needle aspiration) OR (CNB OR core needle biopsy)) AND (axillary lymph nodes OR axillary lymphadenopathy OR axillary staging) AND (Invasive breast cancer OR breast cancer OR breast carcinoma).” Review and non-English language articles were excluded in searching databases. The titles and the abstracts of all searched articles were screened for exclusion. Searched results were then reviewed and articles were included if the study investigated the axillary lymph nodes of breast cancers and there was information for the US-FNAC or CNB. Also, case reports were excluded. The PRISMA checklist is shown in the Appendix A.

### 2.2. Data Extraction

Two individual authors extracted data from all eligible studies. Extracted data from each of the eligible studies included the following [3,4,5,6,7,8,9,10,11,12,13,14,15,16,17,18,19,20,21,22,23,24,25,26,27,28,29,30,31,32,33,34,35,36,37,38,39,40,41,42,43,44,45,46,47,48,49,50,51,52,53,54,55,56,57,58,59,60,61,62,63,64,65,66,67,68,69]: first author’s name, year of publication, study location, number of patients analyzed and the methods of preoperative assessment for ALNs. Besides, for the meta-analysis, we extracted all data associated with the diagnostic accuracy of US-FNAC and CNB in preoperative assessments for ALNs of breast cancers. Numbers of true positive, false positive, false negative and true negative of each method were investigated to obtain the sensitivity, specificity, diagnostic OR and the SROC curve.

### 2.3. Statistical Analysis

To obtain the diagnostic accuracy rate between the US-FNAC and CNB, we performed a meta-analysis using the Comprehensive Meta-Analysis software package 2.0 (Biostat, Englewood, NJ, USA). The diagnostic accuracy rate was evaluated by the concordance between preoperative assessments and histologic diagnosis. Because the eligible studies used various methods for ALNs and had a different number of patients, a random-effects model was more appropriate than a fixed-effects model. Heterogeneity between the eligible studies was checked using *p* statistics (*p*-value). In addition, comparisons between US-FNAC and CNB were performed through a meta-regression test. To evaluate publication bias, we conducted Begg’s funnel plot and Egger’s test. The results with *p* < 0.05 were considered statistically significant. If significant publication bias was found, the fail-safe N and trim-fill tests were additionally conducted to confirm the degree of publication bias. The results were considered statistically significant with *p* < 0.05.

For the DTA review, we used R software ver. 4.0.2 (R Studio, Boston, MA, USA). We calculated the pooled sensitivity and specificity, the diagnostic OR according to individual data, was collected from each eligible study in various categories of comparison. By plotting the ‘sensitivity’ and ‘1-specificity’ of each study, the SROC curve was constructed first and the curve fitting was performed through linear regression. As each dataset was heterogeneous, the accuracy data were pooled by fitting a SROC curve and measuring the value of the AUC. An AUC close to 1 means the test is strong and an AUC close to 0.5 means the test is considered inferior.

## 3. Results

### 3.1. Selection and Characteristics

A total of 330 studies were searched and identified through database searching. Due to insufficient information on concordance rates and diagnostic accuracy, 207 studies were excluded. An additional 45 studies were excluded because they were non-original and 11 were excluded owing to study for other diseases. Finally, 67 studies were included in the present meta-analysis (Figure 1 and Table 1) and they provided data for 11,732 ALNs of breast cancers. Detailed information of eligible studies is shown in Table 1. Various techniques used for US-FNAC and CNB in eligible studies and were described.

### 3.2. Comparison of Diagnostic Accuracy between Fine-Needle Aspiration Cytology and Core Needle Biopsy

The overall diagnostic accuracy for ALNs was 0.850 (95% confidence interval (CI) 0.833–0.866) (Table 2). The diagnostic accuracy of US-FNAC was 0.844 (95% CI 0.825–0.862). In subgroup analysis based on methods of cytological preparation, liquid-based preparation was slightly higher than CS and cell block. The diagnostic accuracy of CNB was 0.896 (95% CI 0.844–0.932). The diagnostic accuracy of CNB was significantly higher than that of US-FNAC (*p* = 0.044 in a meta-regression test).

### 3.3. Diagnostic Test Accuracy Review of Assessments for Axillary Lymph Nodes

Estimated sensitivities of US-FNAC and CNB were 0.760 (95% CI 0.723–0.794) and 0.849 (95% CI 0.776–0.901), respectively (Table 3). In subgroup analysis based on methods of cytological preparation, the sensitivity was the highest in conventional smear than LBP and cell block. Estimated specificities of US-FNAC and CNB were 0.997 (95% CI 0.990–0.999) and 1.000 (95% CI 0.002–1.000). There was no difference in diagnostic OR between US-FNAC and CNB (113.256, 95% CI 71.292–179.922 vs. 119.486, 95% CI 53.021–269.271). Diagnostic OR was the highest in liquid-based preparation compared to other methods. The AUC on SROC of CNB was higher than that of US-FNAC (0.951 vs. 0.922). Diagnostic accuracy of US-FNAC in ALN according to the results of axillary ultrasonography was summarized in Appendix A.

## 4. Discussion

In preoperative assessments of breast cancer, US-FNAC or CNB is recommended for enlarged and suspect in a clinical trial and/or AUS ALNs. In patients found negative using US-FNAC or CNB, SLNB is performed through a frozen biopsy. To reduce unnecessary SLNB, lowering the false-negative rate in US-FNAC or CNB is necessary. If apparent data for preoperative US-FNAC and CNB of ALNs can be obtained, then important information can be further extracted from said data to reduce the false-negative rate. However, at present, there is limited detailed information in individual articles that can be obtained. To the best of our knowledge, the present study is the first meta-analysis and DTA review that compares US-FNAC and CNB for ALN assessment in patients with breast cancer.

Recommendations of preoperative assessment of ALNs can be differed based on AUS findings. If AUS is negative, SLNB without preoperative US-FNAC/CNB is recommended. In patients with suspicious ALNs in AUS, US-FNAC/CNB is recommended to define the preoperative staging. That is, in daily practice, US-FNAC/CNB is performed for only patients with suspicious ALN in AUS. If the LN is judged to be non-suspicious during the AUS, then it is possible that the US-FNAC or CNB did not perform appropriately. Therefore, such cases have no impact on the diagnostic accuracy of US-FNAC/CNB. In addition, when US-FNAC/CNB is performed for non-suspicious ALNs, these cases may be classified as true negatives. In AUS, ALNs are determined to be suspicious or not based on axillary nodal characteristics, such as LN size, cortical thickness, the ratio of long/short axis and a fatty hilum. However, ALNs can also be enlarged by benign conditions, such as hyperplasia and inflammation. If the strict criteria of AUS are applied, the sensitivity of US-FNAC/CNB may increase; however, this may result in an increase in the intraoperative SLNB rate. Therefore, a comparison of the diagnostic accuracy between suspicious and non-suspicious subgroups in AUS would be useful. In the present study, the diagnostic accuracy was significantly higher in the suspicious subgroup than in the non-suspicious subgroup (0.845 vs. 0.726; *p* = 0.048 in a meta-regression test; data not shown). The accuracy of AUS is also important for improving the diagnostic accuracy of preoperative assessments of ALNs.

AUS is a basic and initial diagnostic tool used for patients with breast cancer [70,71]. Although the ability of AUS to provide high-quality images has gradually improved, the appearance of ALNs with normal-appearing morphology ranges between 26% and 52% [8,11,59,72,73,74,75]. The diagnostic accuracy of AUS can be affected by various factors, including the operator’s skill and experience and the ultrasound equipment used. The diagnostic accuracy of preoperative assessments can be improved through US-FNAC/CNB rather than the only US. In a previous study, the false-negative rate was approximately 90% for ALNs smaller than 5 mm. However, following the use of US-FNAC, the false-negative rate decreased to 9–41% [41,76,77]. In the current diagnostic algorithm, US-FNAC was recommended for suspicious ALNs detected during AUS. Lowering the false-negative rate of US and US-FNAC/CNB may be supported by improving the diagnostic accuracy and reducing inappropriate SLNB use.

If tumor cells are identified during US-FNAC or CNB, diagnosis is confirmed as metastatic ALNs. However, when the results of US-FNAC or CNB are negative, the possibility of a false negative by sampling error or overdiagnosis of AUS should be considered [78,79]. Because of these cases, the sensitivity can be lowered. In the pathological evaluation, the tumor foci can be classified into isolated tumor cells, micrometastasis or macrometastasis of ALNs. In preoperative US-FNAC, isolated tumor cells or micrometastasis may result in lower sensitivity compared to macrometastasis [55,74,80,81]. However, the correlation between the size of the metastatic foci and the false-negative rate is not clear. Retrospective confirmation of the diagnostic accuracy of US-FNAC/CNB is not easy because of the challenges presented by targeting and matching ALN-conducted US-FNAC/CNB. In addition, when ALNs with isolated tumor cells or micrometastasis are considered to be non-suspicious findings in AUS, these cases are classified as true negative or skipped US-FNAC/CNB. Thus, ALNs with isolated tumor cells or micrometastasis have no significant impact on the diagnostic accuracy of preoperative US-FNAC/CNB. As per the report of Kane et al. [34], macrometastasis and micrometastasis in the false-negative cases of FNAC were 69% and 31%, respectively. In a previous study, the false-negative rate was significantly correlated with the size of the suspicious ALNs [82]. Furthermore, US-FNAC with inadequate sampling may induce delayed treatments by having to resample [16,83]. On the contrary, the false-positive rates in US-FNAC have been shown to be 1.4–1.7% in previous studies [16,74], while false-positive cases have been shown to be caused by interpretation error during cytological examination [16,74].

US-FNAC has various advantages, including minimal invasiveness, safety, simplicity and low cost. In eligible studies, the sensitivity and specificity of US-FNAC ranged between 0.250 and 0.970 and 0.450 and 1.000, respectively, with a pooled sensitivity of 0.760 (95% CI 0.723–0.794). In daily practice, methods of cytological preparation include CS, LBP and cell block with pooled sensitivities 0.791, 0.784 and 0.643, respectively. The advantage of the cell block method is its ability to conduct ancillary tests. However, the sensitivity of the cell block was shown to be lower than that of the other cytological methods. In daily practice, cell block is additionally prepared with LBP for microscopic examination. Therefore, it would be reasonable to assume that the sensitivity of cell block is similar to the other cytological methods, CS and LBP. The effect of rapid-on site cytologic examination (ROSE) on the assessment of ALNs was investigated in previous studies [20,55]. However, there was no significant difference in the diagnostic accuracy between LBP with and without ROSE. O’Leary et al. reported that ROSE was helpful in assessments of ALNs [55]; however, the false-negative rate did not reduce after the application of ROSE [55]. Although ROSE can improve the sample adequacy, the improvement of diagnostic accuracy is not clear in assessments of ALNs. On the contrary, LBP is an automated method that can conduct ancillary tests, including genetic tests and immunocytochemistry. These ancillary tests may improve diagnostic accuracy; thus, ROSE may be more useful in CS, which is not reproducible.

In CNB, sensitivity and specificity ranged between 0.609 and 1.000 and 0.842 and 1.000, respectively, while the pooled specificities of US-FNAC and CNB were 0.997 and 1.000, respectively. There was no significant difference between the methodology of the preoperative assessments. Some studies reported an improved diagnostic accuracy of CNB compared to US-FNAC [84,85,86]. The sensitivity of CNB was higher than in US-FNAC (0.849 vs. 0.760). Indeed, in the present meta-analysis, the diagnostic accuracy was significantly higher in CNB than US-FNAC (0.896 vs. 0.844; *p* = 0.044 in a meta-regression test). However, we compared the diagnostic accuracy between CNB and each method of cytological preparation. Although a statistical significance between CNB and overall US-FNAC was found, there was no significant difference in the diagnostic accuracy between CNB and each method of cytological preparation.

However, this study has some limitations that need to be addressed. First, the needle gauge size and numbers of passage can be affected by the sample adequacy and diagnostic accuracy. However, a detailed analysis could not be performed due to insufficient information. Second, a detailed analysis for the causes of false-negative rates should be performed in a DTA review. Basically, US-FNAC and CNB have sampling errors; however, because of insufficient information in the eligible studies, detailed analyses could not be conducted. Third, in the previous study, CNB has disadvantages, such as bleeding and high cost, compared to US-FNAC [84]. The technical problem or adverse effect between US-FNAC and CNB will be needed in further studies. In addition, the impact of clinician’s skill on the adequacy of sampling may be more important rather than methodology of biopsy itself. However, this impact could not be evaluated due to insufficient information.

## 5. Conclusions

In conclusion, US-FNAC and CNB are useful diagnostic tools in preoperative assessments of suspicious ALNs in patients with breast cancer. The diagnostic accuracies of various US-FNAC methods are similar and the diagnostic accuracy of CNB is higher than that of US-FNAC.

## Figures and Tables

**Figure 1 diagnostics-10-00717-f001:**
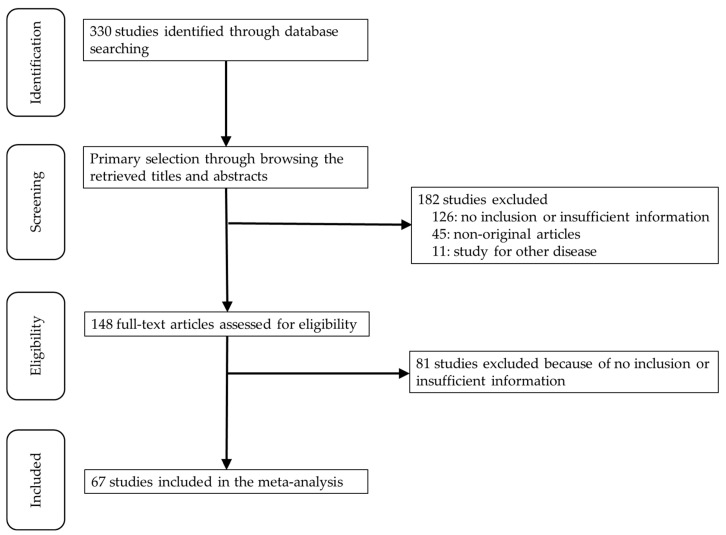
Flow chart of the searching strategy.

**Table 1 diagnostics-10-00717-t001:** Main characteristics of the eligible studies.

Reference	Location	Method	Number of Patients		Reference	Location	Method	Number of Patients	
			Accurate	Total				Accurate	Total
Abe 2009 [3]	USA	CNB (ND)	78	88	Koelliker 2008 [37]	Island	FNAC (LBP)	60	72
Ahn 2013 [4]	Korea	FNAC (CS)	41	48	Kramer 2016 [38]	Netherlands	FNAC (ND)	430	543
		CNB-Stericut	42	48	Krishnamurthy 2002 [39]	USA	FNAC (CS)	75	103
Attieh 2019 [5]	Lebanon	FNAC (ND)	89	101	Kuenen 2003 [40]	Netherlands	FNAC (CS)	103	134
Barco 2017 [6]	Spain	FNAC (ND)	320	390	Leenders 2012 [41]	Netherlands	FNAC (ND)	215	274
Bedrosian 2003 [7]	USA	FNAC (ND)	13	22	Leenders 2013 [42]	Netherlands	FNAC (ND)	363	530
Bonnema 1997 [8]	Netherlands	FNAC (CS)	71	81	Liang 2017 [43]	China	FNAC (ND)	237	263
Boughey 2007 [9]	USA	FNAC (ND)	60	76	Machida 2013 [44]	Japan	FNAC (CS)	33	41
Breitbach 2019 [10]	Germany	FNAC (ND)	46	60	MacNeill 2011 [45]	UK	FNAC (ND)	74	93
		CNB-BARD^®^	10	10	Marti 2012 [46]	USA	FNAC (CS)	78	86
Britton 2009 [11]	UK	CNB-BARD^®^	91	116	Maxwell 2016 [47]	UK	CNB-Achieve^®^	33	37
Bruzzone 2018 [12]	Italy	FNAC (ND)	363	439	Moorman 2015 [48]	Netherlands	FNAC (LBP)	148	202
Caretta-Weyer 2012 [13]	USA	CNB (ND)	24	26	Motomura 2001 [49]	Japan	FNAC (CS)	25	29
Castellano 2014 [14]	Italy	FNAC (CS)	134	146	Nakamura 2018 [50]	Japan	CNB-BARD^®^	260	272
Choi 2015 [15]	Korea	FNAC (CS)	334	373			FNAC (CS)	650	744
Ciatto 2007 [16]	Italy	FNAC (CS)	337	418	O’Leary 2012 [51]	Ireland	FNAC (CS)	108	129
Cools 2013 [17]	Canada	FNAC (ND)	31	53	Park 2011 [52]	Korea	FNAC (CS)	293	382
de Coninck 2016 [18]	Belgium	FNAC (CB)	42	49	Park 2013 [53]	Korea	FNAC (CS)	127	145
de Kanter 2006 [19]	Netherlands	FNAC (ND)	113	161	Podkrajsek 2005 [54]	Slovenia	FNAC (CS)	39	44
Devaraj 2011 [20]	UK	FNAC (ND)	44	45	Popli 2006 [55]	India	FNAC (CS)	20	24
Engohan 2011 [21]	Belgium	FNAC (CB)	19	22	Rao 2009 [56]	USA	FNAC (ND)	18	22
Fayyaz 2019 [22]	Pakistan	FNAC (CS)	136	160			CNB (ND)	21	25
Feng 2015 [23]	China	FNAC (LBP)	1056	1152	Rattay 2012 [57]	UK	FNAC (CS)	49	56
Fung 2014 [24]	USA	FNAC (LBP)	106	130	Rautiainen 2013 [58]	Finland	FNAC (CS)	52	66
García 2011 [25]	Spain	FNAC (CS)	88	96			CNB (ND)	60	66
Genta 2007 [26]	Italy	FNAC (CS)	74	97	Sapino 2003 [59]	Italy	FNAC (CS)	79	85
Gipponi 2016 [27]	Italy	FNAC (ND)	329	400	Schiettecatte 2011 [60]	Belgium	FNAC (LBP)	48	58
Hayes 2011 [28]	Ireland	FNAC (CS)	131	161	Swinson 2009 [61]	UK	FNAC (ND)	87	96
Hyun 2015 [29]	Korea	FNAC (CS)	161	176	Topal 2005 [62]	Turkey	CNB-BARD^®^	36	39
Imai 2018 [30]	Japan	FNAC (CS)	140	162	Tsai 2013 [63]	Taiwan	FNAC (ND)	61	66
Iwamoto 2019 [31]	Japan	FNAC (CS)	140	174	Usmani 2015 [64]	Kuwait	FNAC (LBP)	47	53
Jain 2008 [32]	USA	FNAC (CS)	57	69	Van Berckelaer 2016 [65]	Belgium	FNAC (LBP)	291	317
Jung 2010 [33]	Korea	FNAC (CS)	37	39	Van Wely 2013 [66]	Netherlands	FNAC (CS)	179	198
Kane 2019 [34]	Ireland	FNAC (ND)	480	589	Zhang 2018 [67]	China	FNAC (LBP)	110	124
Kim 2010 [35]	Korea	FNAC (CS)	123	134	Zhong 2018 [68]	China	FNAC (CS)	120	126
Kim 2016 [36]	Korea	FNAC (ND)	24	32	Zhu 2016 [69]	China	FNAC (CS)	235	263

CNB, core needle biopsy; ND, no description; FNAC, fine-needle aspiration cytology; CS, conventional smear; CB, cell block; LBP, liquid-based preparation.

**Table 2 diagnostics-10-00717-t002:** Diagnostic accuracy of ultrasonography-guided fine-needle aspiration cytology and core needle biopsy in the axillary lymph node of breast cancers.

Comparison	Number of Subsets	Heterogeneity (*p*-Value)	Random Effect (95% CI)	Egger’s Test (*p*-Value)	MRT * (*p*-Value)
Preoperative evaluation of ALNs	72	<0.001	0.850 (0.833, 0.866)	0.005	
Fine-needle aspiration cytology	62	<0.001	0.844 (0.825, 0.862)	0.024	0.044
CS	32	<0.001	0.860 (0.839, 0.879)	0.029	0.145
LBP	8	<0.001	0.861 (0.797, 0.908)	0.460	0.332
CB	2	0.942	0.859 (0.758, 0.922)	-	0.544
Core needle biopsy	10	0.002	0.896 (0.844, 0.932)	0.344	

CI, confidence interval; MAR, meta-regression test; ALNs, axillary lymph nodes; CS, conventional smear; LBC, liquid-based preparation; CB, cell block; *, compared to core needle biopsy subgroup in a meta-regression test.

**Table 3 diagnostics-10-00717-t003:** Sensitivity, specificity, diagnostic odds ratio and area under curve of summary receiver operation characteristics curve of ultrasonography-guided fine-needle aspiration cytology and core needle biopsy in axillary lymph node of breast cancers.

Comparison	Included Studies	Sensitivity (%) (95% CI)	Specificity (%) (95% CI)	Diagnostic OR (95% CI)	AUC on SROC
Fine-needle aspiration cytology	62	0.760 (0.723, 0.794)	0.997 (0.990, 0.999)	113.256 (71.292, 179, 922)	0.922
CS	32	0.791 (0.750, 0.827)	0.996 (0.982, 0.999)	122.599 (68.009, 221.008)	0.934
LBP	8	0.784 (0.717, 0.839)	1.000 (0.000, 1.000)	217.586 (49.755, 951.541)	0.917
CB	2	0.643 (0.454, 0.796)	-	72.146 (8.546, 609.058)	0.934
Core needle biopsy	10	0.849 (0.776, 0.901)	1.000 (0.002, 1.000)	119.486 (53.021, 269.271)	0.951

CI, confidence interval; OR, odds ratio; AUC, area under curve; SROC, summary receiver operating characteristic; CS, conventional smear; LBC, liquid-based preparation; CB, cell block.

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
