# Peer review of "Diagnostic Accuracy of Fine-Needle Aspiration Cytology and Core-Needle Biopsy in the Assessment of the Axillary Lymph Nodes in Breast Cancer—A Meta-Analysis"

_diagnostics, 2020, doi:10.3390/diagnostics10090717_

Round 1
Reviewer 1 Report
I suggest a title:
"Diagnostic accuracy of fine-needle aspiration cytology and core-needle biopsy in the assessment of the axillary lymph nodes in breast cancer in women. Meta-analysis. "
Introduction:
- line 55.56; controversial opinion, no literature references
Material and method:
-lack of data on the core needle biopsy technique (e.g. tru-cat) and histopathology examination.
Results:
- in this section we do not evaluate the presented data, there is room for it in the discussion and there is a comparison of our own results with the results of other authors
Discussion:
- line 146; should be ”: US-FNAC or CNB is recommended for enlarged and suspect in a clinical trial and / or AUS ALN.
- Pancreatic cancer topic was touched unnecessarily.
Line 237 - it should be noted that less frequent use of CNB may be justified by the fact that the benefits of CNB do not always exceed the limitations of this method (cost, need for anesthesia, complications)
- One paragraph should be devoted to the limitations of the study, e.g. non-proportional number of studies from US-FNAC and CNB (as mentioned by the authors), only one database was used to search for references.
Conclusions:
-line 245; -without the word "summary"
-line 245; - the statement does not result from work
Author Response
I suggest a title:
"Diagnostic accuracy of fine-needle aspiration cytology and core-needle biopsy in the assessment of the axillary lymph nodes in breast cancer in women. Meta-analysis. "
Response:
As a recommendation, we changed the title.
Introduction:
- line 55.56; controversial opinion, no literature references
Response:
As a recommendation, we added the number of references.
Material and method:
-lack of data on the core needle biopsy technique (e.g. tru-cat) and histopathology examination.
Response:
As a recommendation, we added the information for core-needle biopsy techniques.
Results:
- in this section we do not evaluate the presented data, there is room for it in the discussion and there is a comparison of our own results with the results of other authors
Response:
As pointed out, we described and compared the results of fine-needle aspiration cytology and core-needle biopsy in the result section. However, we aimed to evaluate the diagnostic accuracy between ultrasonography-guided fine-needle aspiration cytology (US-FNAC) and core needle biopsy (CNB) of axillary lymph nodes (ALNs) in patients with breast cancer through a meta-analysis. Therefore, the description for comparisons of results is needed in the result section.
Discussion:
- line 146; should be ”: US-FNAC or CNB is recommended for enlarged and suspect in a clinical trial and / or AUS ALN.
Response:
As a recommendation, we changed the description.
- Pancreatic cancer topic was touched unnecessarily.
Response:
As a recommendation, we deleted the description for pancreatic cancers.
Line 237 - it should be noted that less frequent use of CNB may be justified by the fact that the benefits of CNB do not always exceed the limitations of this method (cost, need for anesthesia, complications)
Response:
We deleted the comments by the recommendation of the reviewer.
- One paragraph should be devoted to the limitations of the study, e.g. non-proportional number of studies from US-FNAC and CNB (as mentioned by the authors), only one database was used to search for references.
Response:
First, we described the limitation of the present study as a separate paragraph (Lines 228-237).
Second, the number of included studies is not included in the limitation of meta-analysis.
Third, we searched using one database (PubMed). However, the number of the database is not in the limitation of meta-analysis.
Conclusions:
-line 245; -without the word "summary"
Response:
The word “summary” is not present.
-line 245; - the statement does not result from work
Response:
As a recommendation, we changed the description as below:
The diagnostic accuracies of various US-FNAC methods are similar, and although the diagnostic accuracy of CNB is higher than that of US-FNAC, the application of US-FNAC or CNB may be impacted by various factors in daily practice.

Reviewer 2 Report
Authors have made a good attempt to review an interesting clinical question. As authors acknowledged, the distribution of FNAC and core biopsy articles are not balanced, as the review includes more of FNAC articles. The main factor in axillary Lymph node biopsy is the degree of skill of the clinician who performs the procedure and adequacy of sampling rather than the method of biopsy itself.
Author Response
Authors have made a good attempt to review an interesting clinical question. As authors acknowledged, the distribution of FNAC and core biopsy articles are not balanced, as the review includes more of FNAC articles. The main factor in axillary Lymph node biopsy is the degree of skill of the clinician who performs the procedure and adequacy of sampling rather than the method of biopsy itself.
Response:
Thank you for the careful review. The comment was added the revised manuscript as below:
In addition, the impact of clinician’s skill on the adequacy of sampling may be more important rather than methodology of biopsy itself. However, this impact could not be evaluated due to insufficient information.

Reviewer 3 Report
- Please define the methods used in selecting the ALNs for CNB/US-FNAC in the evaluated studies- fow was the lymph node selected, if not suspect in ultrasound? Different selection criteria could influence the pooled results
- Please do mention how SLN was selected and defined
- what does ROSE mean?
- please describe the methods compared 0 mention of the technique used for US FNAC and CNB was identical in all considered studies ( including fixation of the smears)
- could you explain the differences in US FNAC in sensitivity ? Pease do commentthe significant values describes in sensitivity. Please do comment the specificity of 100% obtained in CNB results
Author Response
Comments and Suggestions for Authors
1. Please define the methods used in selecting the ALNs for CNB/US-FNAC in the evaluated studies- fow was the lymph node selected, if not suspect in ultrasound? Different selection criteria could influence the pooled results
Response:
We uploaded supplementary Table 2, which has a comparison of diagnostic accuracy between suspicious and non-suspicious subgroups in AUS. However, the suspicion of axillary ultrasound sonography (AUS) was not included in the selection criteria. Therefore, the pooled results were not affected by the selection criteria.
2. Please do mention how SLN was selected and defined
Response:
Data for sentinel lymph node (SLN) biopsy was not investigated in the present meta-analysis. Therefore, we could not describe the selection and definition of SLN in the present meta-analysis.
3. what does ROSE mean?
Response:
The full-name of “ROSE” is a rapid-on site cytologic examination. We added the full name in the revised manuscript.
4. please describe the methods compared 0 mention of the technique used for US FNAC and CNB was identical in all considered studies (including fixation of the smears)
Response:
As a recommendation, we added the comment in the revised manuscript.
5. could you explain the differences in US FNAC in sensitivity?
Response:
The sensitivities of US-FNAC and CNB were respectively 0.760 and 0.849. In addition, sensitivities were different between the methodology of US-FNAC. The aim of a meta-analysis is to obtain the integration of the results from previous studies. The various factors may affect the sensitivity. However, we could not be found the cause of different sensitivities in the results of the meta-analysis.
Please do comment the significant values describes in sensitivity.
Response:
We added the comment in the discussion part of the revised manuscript.
Please do comment the specificity of 100% obtained in CNB results
Response:
We previously described in the manuscript.
